# Cluster Topology-Driven Placement of Experts Reduces Network Traffic in MoE Inference

## Abstract

Efficient deployment of a pre-trained LLM to a cluster with multiple servers is a critical step for providing fast responses to users' queries. The recent success of Mixture-of-Experts (MoE) LLMs raises the question of how to deploy them efficiently, considering their underlying structure. During the inference in MoE LLMs, only a small part of the experts is selected to process a given token. Moreover, in practice, the experts' load is highly imbalanced. For efficient deployment, one has to distribute the model across a large number of servers using a model placement algorithm. Thus, to improve cluster utilization, the model placement algorithm has to take into account the network topology. This work focuses on the efficient topology-aware placement of the pre-trained MoE LLMs in the inference stage. We propose an integer linear program (ILP) that determines the optimal placement of experts, minimizing the expected number of transmissions. Due to the internal structure, this optimization problem can be solved with a standard ILP solver. We demonstrate that ILP-based placement strategy yields lower network traffic than competitors for small-scale (DeepSeekMoE 16B) and large-scale (DeepSeek-R1 671B) models.

## 1 Introduction

This paper proposes a novel ILP-based framework for placing the Mixture of Experts (MoE) transformer model (Shazeer et al., 2017; Lepikhin et al., 2020; Fedus et al., 2022) over diverse cluster topologies. We consider cluster topology as an undirected graph, where vertices correspond to GPUs on servers and edges are direct links between GPUs and servers. The granularity of the connections between GPUs from the same server and the connections between different servers is modeled with edge weights. In particular, the edges between GPUs on the same server have zero weights due to the extremely fast interconnect between GPUs. MoE transformer model is a modification of the classic transformer model, where a linear layer after the attention block is replaced by a dynamically routed set of linear layers called *experts*. This modification of the classic transformer model leads to better performance (Fedus et al., 2022). Although the necessary VRAM increases, the number of loaded experts during every token processing in the inference stage remains limited. Therefore, the latency of the MoE transformer model is comparable to that of much smaller and less accurate models (Fedus et al., 2022). Since a limited number of experts is used to process each token, a large batch size is used to increase the utilization of GPUs by each expert. Thus, the increasing VRAM and large batch size require multiple GPUs and, thus, servers for efficient deployment of the MoE transformer model. At the same time, the imbalanced loading of expert layers (Zhang et al., 2022) and distributed setup make the proper placement of expert layers crucial for efficient utilization of available GPUs by the entire MoE transformer model. Our framework optimizes the placement of the MoE transformer model on GPUs, ensuring that the path length from the highly loaded experts to the previous and subsequent attention layers is minimized.

The problem of placing deep learning models in the cluster is not novel (Verbraeken et al., 2020). However, the primary purpose of standard placement approaches is to enhance the GPUs' efficiency during the training stage (Gusak et al., 2022) since the inference task in the distributed setup was not challenging for non-MoE-based models. In contrast, inference performance in MoE transformer models is more sensitive to the placement of experts, and proper placement can significantly enhance

cluster utilization. The known underlying structure of such models leads to a specific, tractable ILP problem. The ILP framework was also used in MoETuner (Go & Mahajan, 2025), which aims to balance the load of experts. Additionally, the authors considered only one or two servers and assumed that the network topology is represented as a complete graph, which is an infeasible assumption for a multi-server cluster. In this work, we also utilize the statistics of the experts' load, which makes our framework more robust to an imbalanced distribution of experts' loads.

The main contributions of our work are the following:

- We formalize the problem of placing the MoE layers over a cluster using ILP framework.
- We demonstrate that exploiting statistics of experts' load in the optimization problem significantly improves the placement of experts.
- We empirically confirm that the proposed ILP-based placement strategy yields lower network traffic for the DeepSeek-MoE 16B and DeepSeek-R1 671B models across four different network topologies.

## 2 RELATED WORKS

The efficient deployment of the pre-trained model remains a challenging task due to the significant increase in model size and architectural features. Further in this section, we describe the main directions used to make the efficient deployment only for the inference stage.

**Cluster topology.** The naïve approach to managing the topology of the computational cluster is to connect every server with all others. This approach corresponds to the complete graph, where vertices are servers and edges are links between servers. However, it is not scalable and costs a lot compared to custom sparse topologies (Hoefler et al., 2024). To reduce the cluster construction costs, numerous topologies were proposed, e.g., FatTree (or Clos) (Singh et al., 2015), Dragonfly (Kim et al., 2008), Dragonfly+ (Shpiner et al., 2017), Slim Fly (Besta & Hoefler, 2014), and others (Hoefler et al., 2024). Most of them were initially designed for general-purpose Ethernet clusters, rather than AI applications. In addition, general routing algorithms (Besta et al., 2020) and efficient collectives implementations (Prisacari et al., 2013; Zahavi, 2012) are proposed, while they still ignore issues raised in the inference stage. Such issues arise only for large MoE LLMs and have not been so viable for the previous models. Therefore, the architecture of deep learning models, and in particular MoE LLMs, can mismatch the cluster topology, making heuristic placement algorithms inefficient. Thus, the specific approaches for efficient placement of the MoE LLMs in the clusters with a given topology are necessary for the successful deployment of such models in services.

**Expert placement.** There are two complementary stages of expert placement. First, *initial placement* distributes experts across GPUs before request processing begins. Second, *adaptive balancing* involves replicating hot or imbalanced experts on additional GPUs to absorb spikes in activation frequency. Initial placement targets expert imbalance arising from imperfect training (Shazeer et al., 2017) and from data shifts coming from the deployment for domain-specific tasks. MoETuner (Go & Mahajan, 2025) performs initial placement by formulating an ILP problem that balances expert load. We show in Table 1 that using the same objective at the cluster scale leads to an intractable optimization problem that can not be used in practice. In this paper, we address the initial placement problem. However, instead of directly optimizing load equality as MoETuner does, we minimize the datacenter traffic amount subject to balancing constraints.

At the same time, adaptive balancing of experts' load complements the initial placement of experts. Lynx (Gupta et al., 2024) observes that production servers batch requests, which can activate many experts. To address this problem, the authors propose dynamic, batch-aware expert selection to shrink the active expert set per batch and reduce decode-phase latency. Complementary techniques, such as dynamic gating, expert buffering, and expert load balancing, are outlined in (Huang et al., 2024). Balmau et al. (2025) propose sharding expert matrices to distribute the load evenly across GPUs. However, the induced tensor-parallel execution becomes increasingly network-bound as the degree of parallelism increases. Finally, works such as (Guo et al., 2024) modify the gating mechanism itself to better align with parallel execution. The proper combination of the proposed approach to initial placement and adaptive balancing methods is the topic for future work.

# 3 MODEL PLACEMENT PROBLEM

This section presents the considered expert placement problem for MoE models and discusses why this problem appears to be actual and relevant for this class of Transformer models. In the inference stage, networking issues were not typically a significant concern, as inference time scaling techniques like pipelining are typically implemented within a single server or a tightly clustered group. Therefore, the deployment can be scaled by the number of such groups. However, there is a different situation with MoE models since the experts' activation per token appears to be very sparse, i.e., only a few experts process every token. Based on the theoretical analysis, a MoE model may have a very small memory footprint in inference (up to 3-10% of all model parameters). However, in practice, there are several challenges:

1. Most computations performed by experts consist of matrix multiplications. This compute-bound operation requires a large number of matrices to utilize GPUs efficiently.

2. Although MoE models are trained to ensure a balanced load between experts, experts are unevenly balanced in practice, e.g., some are activated twice as often as others.

3. Dynamic experts routing per token leads to an impossible prior placement of experts since one can not know which will be picked until per-layer router activations are computed. Therefore, any static placement of experts may lead to poor GPU utilization.

4. The recent pretrained MoE models (Guo et al., 2025; Dai et al., 2024; Team et al., 2025) are designed to have a large number of total parameters. Therefore, if the model can even fit on a single server, there may not be sufficient VRAM to store KVCache activations.

A possible solution to deal with the first three challenges is significantly increasing the load per model instance. In this case, each expert will be statistically loaded enough to achieve efficiency in performing the individual expert's matrix multiplication. However, this approach leads to high memory demand for one deployment unit. To handle such a load per unit, one needs to support a high parallelization degree over dimensions of weights and experts. For example, the Deepseek team reported using at least 320 GPU inference pods with 320 degree parallelism of experts (256 GPUs for each unique expert and 64 redundancy experts).

Another thing about inference is that all cross-GPU communications can be considered as point-to-point, since there is only token dispatch and collect network communication for parallelized model state. This is partially applicable for training, except that the All-Reduce operation for gradient averaging across servers typically uses complicated implementations to deal with network topologies and congestion efficiently.

Thus, the experts' placement of the pre-trained large-scale MoE models has to consider the topology of the network and expectations on the experts' loading. Further, we propose the ILP-based approach that considers both these ingredients. However, to smoothly introduce the reader to our approach, we briefly describe the necessary concepts and provide the key notations.

## 3.1 NETWORK SETUP

Let $G_n = (V_n, E_n)$ be a graph of the network, where the vertices are computational servers and the edges are cross-server connections. Denote by $S = |V_n|$ the number of servers and by $n = |E_n|$ the number of edges. There are multiple classic topologies for the cluster network configurations, like Fat-Tree (Clos) (Singh et al., 2015), Slim Fly (Besta & Hoefler, 2014), Dragonfly (Kim et al., 2008), Dragonfly+ (Shpiner et al., 2017). However, we consider FatTree (Singh et al., 2015) and Dragonfly (Kim et al., 2008) topologies as the most representative ones. The visualizations corresponding to these topologies are presented in Figures 1a and 1b.

One of the main features of the topologies is the length of the shortest path between two arbitrary servers. To illustrate this feature for the FatTree and DragonFly topologies, we present Figure 2, where the difference in the topologies is represented through the corresponding pairwise distance matrices. So, the FatTree corresponds to the block diagonal pairwise distance matrix, and the DragonFly corresponds to the same block diagonal matrix plus a grid of non-diagonal elements.

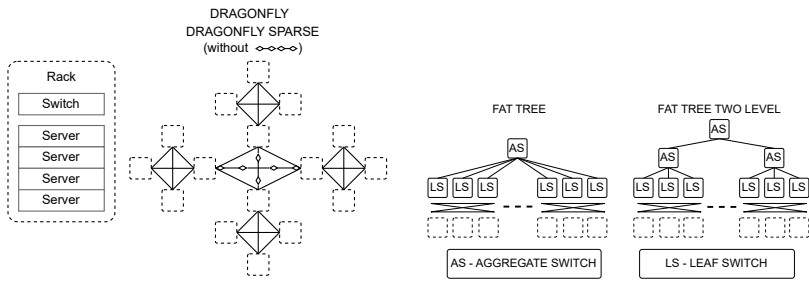

(a) Dragonfly and Dragonfly Sparse topologies, with rack notation.

(b) Fat Tree and two-level Fat Tree topologies, with dotted boxes denoting racks (see Fig. 1a).

Figure 1: Comparison of network topologies.

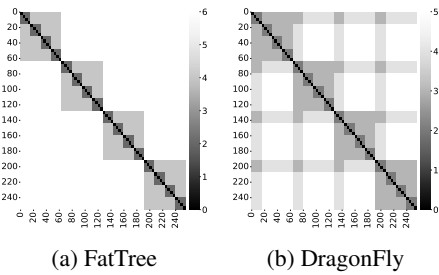

(a) FatTree

(b) DragonFly

Figure 2: Pairwise distance heatmaps with the lengths of shortest paths between every two servers for the considered topologies. Each topology is built over 256 GPUs, with four servers per rack and 4 GPUs per server. Distances inside the server are zero because a fast interconnect is assumed.

## 3.2 MODEL SETUP

To demonstrate the proof of concept and scalability of the proposed approach, we consider DeepSeekMoE 16B (Dai et al., 2024) and DeepSeek-R1 (Guo et al., 2025) models. In Mixture-of-Experts (MoE) models, a standard Feedforward block (FFN) with linear layers and activations is replaced by a MoE block with many dynamically routed experts. Experts have an identical structure and size as FFN blocks with different weights and transform the input for the $t$-th token $\mathbf{u}_t$ to the output $\mathbf{h}_t$ according to the following equation:

$$\mathbf{h}_t = \mathbf{u}_t + \sum_{i=1}^{N_s} \mathrm{FFN}_i^{(s)}(\mathbf{u}_t) + \sum_{j=1}^{N_r} g_{jt}\,\mathrm{FFN}_j^{(r)}(\mathbf{u}_t), \tag{1}$$

$$g_{jt} = \frac{g'_{jt}}{\sum_{k=1}^{N_r} g'_{kt}}, \; g'_{kt} = \begin{cases} s_{kt} & s_{kt} \in \underset{1 \le k \le N_r}{\mathrm{Top\,K_r}}\left(\{s_{kt}\}\right) \\ 0 & \text{otherwise}, \end{cases} \tag{2}$$

$$s_{kt} = \sigma\left(\mathbf{u}_t^\top \mathbf{e}_k\right), \tag{3}$$

where the $N_s$ *shared* experts $\mathrm{FFN}_i^{(s)}$, $i = 1, \ldots, N_s$ and the $N_r$ *routed* experts $\mathrm{FFN}_j^{(r)}$, $j = 1 \ldots, N_r$ are processed input embedding $\mathbf{u}_t$. The routed experts are controlled by the normalized gate $g_{jt}$ such that it is nonzero only for the top-$K_r$ routed experts selected by the FFN router with trainable weights $\mathbf{e}_k$. Further, we denote by $L$ the number of transformer blocks, containing MoE layers. We also consider models where $N_s = 1$, and to simplify notations, we denote the number of routed experts per MoE layer by $E := N_r$.

The experts' placement problem arises explicitly from the dynamical router outputs and, thus, the selection of routed experts. The imbalanced loading of experts for the DeepSeek-R1 model is shown in Figures 3a and 3b and confirms the practical importance of the proper experts' placement for such MoE models. This feature of the MoE transformer model motivates the development of the specific placement algorithm for the inference stage.

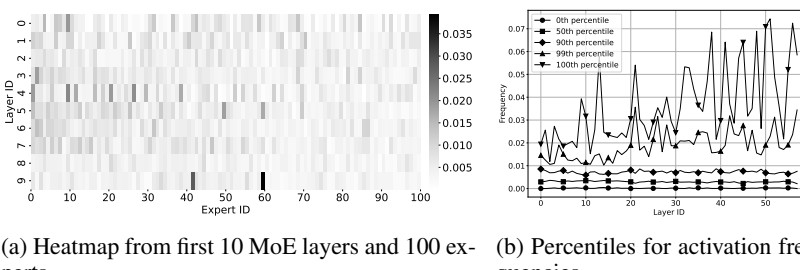

(a) Heatmap from first 10 MoE layers and 100 experts

(b) Percentiles for activation frequencies

Figure 3: Imbalanced expert load in inference for DeepSeek-R1 model and OASST1 dataset.

**Objective function.** We optimize the average number of network hops for processed tokens. A hop is a single point-to-point activation communication between two servers connected by a link. Each token goes through all shared layers of models (mostly, attentions) and an individual subset of routed experts on each MoE layer. The objective function averages hops over all such communications. Each logical communication is translated into a sequence of point-to-point communications, so the hop corresponds to the length of the OSPF (Moy, 1998) path between the source and destination servers. The formalization of the introduced objective function is presented in Section 4.3.

**Constraints.** To balance the distribution of experts between servers, we introduce the following constraints. The first constraint forces that only a single expert is placed on the server. This constraint prevents the under-utilization of the available resources. The second constraint limits the total number of experts that a server can store by $C_{\mathrm{exp}}$. The third constraint limits the total number of experts *from one layer* that a server can store by $C_{\mathrm{layer}}$. This constraint prevents unequal load distribution, high peak GPU memory consumption, and imbalanced KVCache during inference. The formal definitions of these constraints are presented in Section 4.3.

## 4 METHODS

This section describes the proposed ILP-based placement algorithms and the baselines. In particular, we consider the Round-Robin Buyya et al. (1999) (RR) and greedy placement algorithms since they are the standard options that could be used in practice and perform the model placement fast.

### 4.1 ROUND ROBIN (RR) PLACEMENT.

The first baseline is the classical Round Robin algorithm adjusted for our domain. This adjustment is performed in the following way. Firstly, the available GPUs in servers are enumerated sequentially such that the closer GPUs to each other according to the shortest path length (see Figure 2), the closer their indices. After that, for every attention layer, we take the index $i$ of the GPU, where it is stored, and place the experts, following this attention, on GPUs corresponding to the $d/2$ left and $d/2$ right indices, where $d = \frac{E}{C_{\mathrm{layer}}}$ and $E$ is the number of routed experts per MoE layer. Since the considered topologies are symmetric, the sorted list of GPU indices can be represented as a circle, so the boundary effects if $k$ is small or large are avoided. This approach leads to tight packing of experts to GPUs and aims to preserve locality for the dispatch attention layer. However, RR fails to capture the distance to the collect attention layer, which uses the experts' outputs as input for further processing. This drawback is addressed by the Greedy approach, presented in the following section.

### 4.2 GREEDY PLACEMENT.

A more complicated strategy is the Greedy approach. This approach also starts from the enumeration of available GPUs, similar to RR. However, the next step significantly differs. For every layer $\ell$ and every expert $e$, the indices of GPUs are sorted in ascending order according to the following key $\mathrm{dist}(d_\ell, s) + \mathrm{dist}(s, c_\ell)$, where $d_\ell$ and $c_\ell$ correspond to the current (dispatch) and next (collect) attention layers and $s$ denotes the GPU's index. After such sorting, the experts are placed on a GPU such that its index in the sorted list is the smallest and the placement constraints are satisfied.

We expect that the Greedy approach dominates RR since it takes into account distances to both dispatch and collect attention layers. However, it has two natural drawbacks: the greedy approach does not necessarily provide the optimal solution for the entire placement of all experts, and it ignores the statistics of experts' loading, which is typically imbalanced (see Figures 3a and 3b) and affects the proper placement.

### 4.3 INTEGER LINEAR PROGRAMMING PLACEMENT.

Although the heuristics mentioned above are straightforward, they do not guarantee the optimality of the placement. In contrast, the 0–1 Integer Linear Programming approach explicitly optimizes the introduced objective function and provides the *optimal* placement of experts for a given network topology and pre-defined model setup. To formalize the placement of experts in MoE model, we introduce the binary variables $y_{\ell es} \in \{0, 1\}$, where $\ell \in \{1, \ldots, L\}$, $e \in \{1, \ldots, E\}$ and $s \in \{1, \ldots, S\}$. The interpretation of these variables is the following: $y_{\ell es} = 1$ iff the expert $e$ corresponding to the MoE layer $\ell$ is put on the server $s$. Then, the straightforward objective function is the number of network hops per forward pass. This quantity can be computed as

$$\sum_{\ell=1}^{L}\sum_{e=1}^{E}\sum_{s=1}^{S} p_{\ell s} y_{\ell es},$$

where $p_{\ell s} = \mathrm{dist}(d_\ell, s) + \mathrm{dist}(s, c_\ell)$ and $\mathrm{dist}(s, c_\ell)$ denotes the length of the shortest path from $s$ and $c_\ell$. We denote by $d_\ell$ and $c_\ell$ the indices of attention blocks corresponding to layers $\ell$ that dispatch and collect experts, respectively. Thus, taking into account the constraints from, we can state the following integer linear optimization problem:

$$\min_y \sum_{\ell=1}^{L}\sum_{e=1}^{E}\sum_{s=1}^{S} p_{\ell s} y_{\ell es} \quad \text{s.t.} \quad \sum_{s=1}^{S} y_{\ell es} = 1, \forall \ell \in \mathcal{L}, \ e = 0, \ldots, E-1$$

$$\sum_{\ell=1}^{L}\sum_{e=0}^{E-1} y_{\ell es} \leq C_{\mathrm{exp}}, \forall s \in \mathcal{S} \qquad \sum_{e=0}^{E-1} y_{\ell es} \leq C_{\mathrm{layer}}, \forall \ell \in \mathcal{L}, \ s \in \mathcal{S} \tag{4}$$

The problem (4) provides the optimal solution that takes into account only the distance between servers. The placement given by the solution of problem (4) is further referred to as `ILP`. At the same time, according to Figure 3b, the load of experts is not uniform, which directly affects the utilization of available hardware. To avoid a utilization drop and include the prior knowledge on expected experts' load, we modify the objective function from (4) and propose to use the experts' load frequencies estimated from a reference dataset. We use the OASST1 dataset (Köpf et al., 2023) and provide more details about it in Section 5. Denote by $f_{\ell e}$ the frequency of loading expert $e$ from layer $\ell$, so $\sum_{e=1}^{E} f_{\ell e} = 1$ for every layer $\ell$. Then, the load-aware objective function looks as follows

$$\sum_{\ell=1}^{L}\sum_{e=1}^{E}\sum_{s=1}^{S} f_{\ell e} p_{\ell s} y_{\ell es}$$

and could be interpreted as the expected load of experts. The resulting optimization problem is composed with this objective function and the constraints from (4). The placement obtained from the load-aware optimization problem is further referred to as `ILPLoad`.

Table 1 provides the summary of the considered methods with the type of the solution and the runtime to obtain it for the DeepSeek-16b model. We observe that heuristic methods work very fast but provide only sub-optimal solutions. At the same time, our approaches (ILP and ILPLoad) give the optimal solutions and require much less runtime than MOETuner. The runtime of ILP and ILPLoad does not prevent the use of these approaches in practice since the initial placement of experts is performed for a sufficiently long period of operating the queries.

## 5 EXPERIMENTS

In this section, we show the gain from the proposed ILP framework on two MoE models (DeepSeek-MoE 16B and DeepSeek-R1 671B) and four cluster topologies.

Table 1: Summary of the presented methods. * - timeout after 12 hours, even on the toy DeepSeek-16b model. The proposed ILP and ILPLoad provide exact solutions while requiring reasonable runtime compared to MOETuner.

| Method | Exact solution | Runtime, s |
|---|---|---|
| MOETuner | Yes | timeout* |
| Round-robin | No | 0.19 |
| Greedy | No | 0.79 |
| ILP | Yes | 1185.9 |
| ILPLoad | Yes | 1397.5 |

## 5.1 Test cluster configuration.

We evaluate the performance of the proposed ILP and ILPLoad approaches for the following cluster configuration. The total number of GPUs is 256, and each server has 4 GPUs. Each leaf switch is connected to 4 servers; we have 16 leaf switches. Leaf switches are connected in the following topologies: Dragonfly, Fat-Tree, Sparse Dragonfly, and hierarchical Fat-Tree. If two GPUs reside on the same server, we assume a distance of 0 between them, since NVLink has a higher bandwidth.

## 5.2 Performance of the ILP framework.

We evaluate the performance of the placement algorithm on real statistics collected from the DeepSeek-R1 MoE model with 256 routed experts per layer (8 experts are loaded per token). To collect statistics, we used the OASST1 dataset (Köpf et al., 2023). The evaluation metric is the average number of network hops in the corresponding network topologies for the described model. We use a disjoint subset of activations from the OASST1 dataset as a dataset for activations.

Table 2 shows that only ILPLoad leads to significantly better placement of experts than competitors if the number of racks is large. A large number of racks makes the influence of the topology on the placement quality dominant. Therefore, this setup demonstrates that since ILPLoad takes into account both network topology and load statistics, it outperforms the competitors. Moreover, these results indicate that if we increase the total number of GPUs, we can expect a more significant gain.

Table 2: Only ILPLoad performs significantly better than Round-Robin (RR), if we model a very diverse cluster with 64 servers, each equipped with 1 GPU, and one server per rack. For this artificial setup, we use statistics from the DeepSeek-MoE 16B model with 27 layers, each with 64 experts. We use weak constraint $C_{\text{layer}} = 1$.

| Network | Placement | Hops | Gain | Network | Placement | Hops | Gain |
|---|---|---|---|---|---|---|---|
| FatTree | RR | 1819.75±6.70 | | FatTree | RR | 2148.93±11.36 | |
| | Greedy | 1815.80±11.41 | 0.2% | Sparse | Greedy | 2137.15±16.60 | 0.6% |
| | ILP | 1827.71±7.91 | -0.4% | | ILP | 2154.07±11.89 | -0.2% |
| | ILPLoad | **1750.95**±43.15 | **3.9%** | | ILPLoad | **1999.03**±77.34 | **7.5%** |
| Dragonfly | RR | 1366.48±6.24 | | Dragonfly | RR | 1736.71±14.37 | |
| | Greedy | 1364.06±9.93 | 0.2% | Sparse | Greedy | 1724.76±14.77 | 0.7% |
| | ILP | 1367.39±6.81 | -0.1% | | ILP | 1732.91±10.33 | 0.2% |
| | ILPLoad | **1289.53**±41.24 | **6.0%** | | ILPLoad | **1574.85**±79.58 | **10.3%** |

Now, we consider the DeepSeek-R1 model, and the constraint $C_{\text{layer}} = 1$ is taken from tech report (Liu et al., 2024). Table 3a shows significant degradation of the basic ILP, yet ILPLoad outperforms competitors. Note that ILPLoad has the largest standard deviation because prioritizing experts' usage results in short paths for commonly used experts and long paths for rare experts.

The experimental results presented in Table 3b correspond to the relaxing constraints on $C_{\text{layer}}$ value, setting it $C_{\text{layer}} = 8$, while preserving the MoE model and other cluster parameters. In this scenario, ILPLoad still provides the best placement of experts. In addition, note that the basic ILP approach gives worse placement than the Greedy algorithm. The analysis of the reason for this observation is the topic of future work.

Table 3: ILPLoad is the best among all topologies with DeepSeek-R1 inference pod's like ($C_{\text{layer}} = 1$) and with relaxed ($C_{\text{layer}} = 8$) constraints. Performance comparison for different $C_{\text{layer}}$. Each topology is displayed with a real-world cluster model, featuring four servers per rack, 4 GPUs per server, and assumed GPUs interconnect usage.

(a) ILPLoad is the best among all topologies with DeepSeek-R1 inference pod's like constraints ($C_{\text{layer}} = 1$). Greedy gives moderate gain, and basic ILP performs close to Round-Robin.

| Network | Placement | Hops | Gain |
|---|---|---|---|
| FatTree | RR | 5003.98±19.72 | |
| | Greedy | 4755.52±48.03 | 5.2% |
| | ILP | 4952.75±27.12 | 1.0% |
| | ILPLoad | **4391.73**±186.00 | **13.9%** |
| Dragonfly | RR | 3757.23±14.78 | |
| | Greedy | 3561.30±34.66 | 5.5% |
| | ILP | 3699.78±18.43 | 1.6% |
| | ILPLoad | **3280.58**±140.82 | **14.5%** |
| FatTree Sparse | RR | 5980.81±33.91 | |
| | Greedy | 5547.00±66.66 | 7.8% |
| | ILP | 5896.66±49.12 | 1.4% |
| | ILPLoad | **4995.65**±273.06 | **19.7%** |
| Dragonfly Sparse | RR | 4009.85±20.05 | |
| | Greedy | 3757.44±41.46 | 6.7% |
| | ILP | 3935.40±24.67 | 1.9% |
| | ILPLoad | **3421.65**±160.87 | **17.2%** |

(b) ILPLoad performs consistently better even with relaxing constraints ($C_{\text{layer}} = 8$). ILP provides a minor gain compared to the RR. The Greedy algorithm is still the second-best.

| Network | Placement | Hops | Gain |
|---|---|---|---|
| FatTree | RR | 2872.62±20.92 | |
| | Greedy | 2426.58±30.08 | 18.4% |
| | ILP | 2649.11±18.51 | 8.4% |
| | ILPLoad | **2198.12**±117.68 | **30.7%** |
| Dragonfly | RR | 2258.62±13.18 | |
| | Greedy | 1990.23±21.82 | 13.5% |
| | ILP | 2151.10±17.17 | 5.0% |
| | ILPLoad | **1826.13**±84.02 | **23.7%** |
| FatTree Sparse | RR | 2992.34±22.46 | |
| | Greedy | 2442.58±30.08 | 22.5% |
| | ILP | 2643.34±16.22 | 13.2% |
| | ILPLoad | **2229.95**±117.47 | **34.2%** |
| Dragonfly Sparse | RR | 2258.62±13.18 | |
| | Greedy | 1990.23±21.82 | 13.5% |
| | ILP | 2128.97±10.70 | 6.1% |
| | ILPLoad | **1826.66**±83.97 | **23.6%** |

**Ablation study of $C_{\text{layer}}$ values.** This paragraph presents how the value of $C_{\text{layer}}$ affects the performance of the considered methods while other cluster parameters are the same as in previous experiments. We expect that the larger the $C_{\text{layer}}$ is, the smaller the gap between the approaches is. The reason for this expectation is that a large $C_{\text{layer}}$ enables tight packing of experts by all algorithms. Figure 4a shows the dependence of ILPLoad and Greedy placements quality on the $C_{\text{layer}}$. We begin an ablation study using these algorithms, as they are the two best methods from the previous experiments. Note that increasing $C_{\text{layer}}$ leads to a lower relative difference between algorithms. Lower $C_{\text{layer}}$ leads to higher variance, as unfrequent experts are placed much further from their dispatch and collect servers.

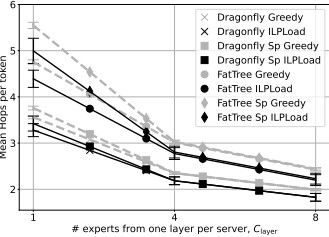 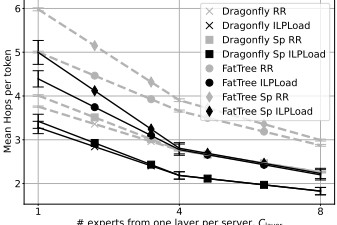 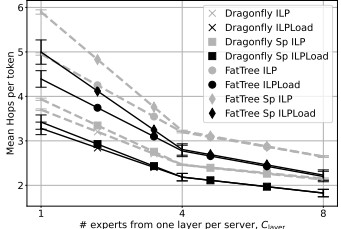

(a) The gap between ILPLoad and Greedy narrows as $C_{\text{layer}}$ increases.

(b) ILPLoad outperforms RR uniformly while RR shows lower variance.

(c) ILPLoad dominates ILP in terms of mean while providing larger variance.

Figure 4: Dependence of the average numbers of hops on the $C_{\text{layer}}$. The gap between algorithms is lower, while the $C_{\text{layer}}$ becomes larger.

Figure 4b shows the difference between ILPLoad and Round-Robin placements when $C_{\text{layer}}$ is changing. Note that increasing $C_{\text{layer}}$ leads to a lower relative difference between algorithms, and Round-Robin has much lower variance than ILPLoad. Figure 4c shows the difference between ILPLoad and ILP placements when $C_{\text{layer}}$ is changing. Note that increasing $C_{\text{layer}}$ leads to a lower relative difference between algorithms and ILP, as others have much lower variance than ILPLoad.

### 5.3 INTERPRETATION OF THE ILP-GENERATED PLACEMENT.

This section presents how the ILPLoad-generated placement differs from the second-best approach for Dragonfly and FatTree topologies. In particular, Figure 5 shows that for both topologies the ILPLoad adjusts the experts' placement according to the distance matrices presented in Figure 2. This observation is especially clear from Figure 5b, where many non-zero elements out of the block diagonal structure are aligned with the similar structure of the distance matrix in Figure 2b.

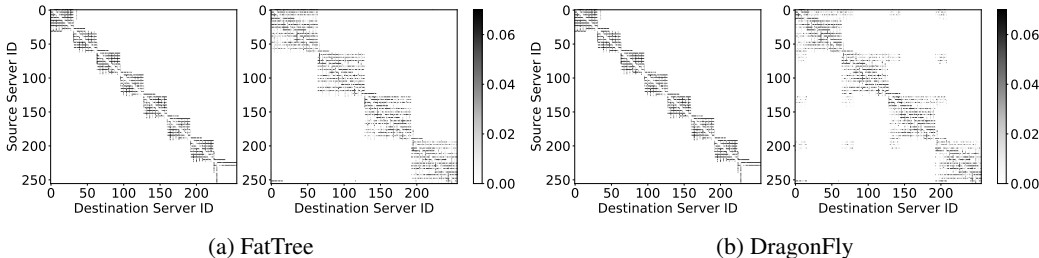

(a) FatTree                                      (b) DragonFly

Figure 5: Expert's frequency weighted communication map for two best placements for different topologies: left - Greedy, right - ILPLoad. DeepSeekR1 model statistics and $C_{\text{layer}} = 8$ are used.

### 5.4 DRIFT OF THE EXPERTS LOADING

Computing an efficient placement does not allow us to rebalance experts in real time. For this reason, we study how dataset drift influences the final results. Table 4 shows that even when we train on one dataset and evaluate on a different one, the performance of ILPLoad remains stable and consistently improves over the initial round-robin placement. In practice, traffic drift is typically smoother, so our algorithm can be recomputed periodically using statistics collected over a recent time window.

Table 4: Total number of hops when training ILPLoad on statistics from the dataset in the row and validating on the dataset in the column. In parentheses, we report the percentage improvement of ILPLoad over the initial round-robin placement. Experiments use a 16B model and a $C = 2$ scenario on a fattree sparse topology.

|        | osst          | alpaca        | vicuna        |
|--------|---------------|---------------|---------------|
| osst   | 1999.0 (+7%)  | 2083.9 (+3%)  | 2097.6 (+2%)  |
| alpaca | 2087.8 (+3%)  | 2029.5 (+6%)  | 2084.4 (+3%)  |
| vicuna | 2079.8 (+3%)  | 2065.1 (+4%)  | 2063.3 (+4%)  |

Figure 6: Expert activations distributions for different datasets. We observe that the distribution of the expert activations is not very far, which aligns with the observed difference in Table 4.

## 6 CONCLUSION

This study presents an ILP-based framework for the optimal placement of MoE LLM models within a cluster to reduce the traffic load between servers during the inference regime. The reduction of traffic load is crucial for the deployment of LLM models to generate responses to users' queries efficiently. We consider several of the most popular topologies of clusters and pre-trained advanced MoE LLM models. To minimize the traffic load for inner communications between servers during inference in the considered models, we state the ILP problem such that its solution corresponds to the optimal placement of MoE LLM over the available servers. The experimental comparison of our approach and standard heuristics demonstrates that the ILP-based framework provides better utilization of the cluster and minimizes communication overhead. The moderate runtime for solving the target ILP problem makes our approach practically essential and relevant for the industry.

## LIMITATIONS

The most significant limitation of the presented work is the lack of access to the standalone cluster, which would enable us to benchmark the network load reduction for the compared configurations of the MoE LLM models. Another limitation is the assumptions used in the ILP problem that focus on the high-level distributions of experts and attention layers only.

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

## A APPENDIX

### A.1 ADDITIONAL EXPERIMENTS

For completeness, we present an additional table with raw measurements used in $C_{\text{layer}}$ ablation. In main text, there are results for $C_{\text{layer}} = 1$ in Table 3a, and for $C_{\text{layer}} = 8$ - Table 3b. Table 5 completes raw measurements, displayed in ablation Figure 4a- 4c, by showing measurements for $C_{\text{layer}} = 4$. ILP Load, as in previous comparisons, is best among the compared algorithms.

Table 5: ILPLoad is the best among all topologies with $C_{\text{layer}} = 4$. Greedy gives gain closer to ILPLoad, yet still lose up 10%, and basic ILP performs in the middle between Greedy and Round-Robin placement. Each topology is displayed with a real-world cluster model, featuring four servers per rack, 4 GPUs per server, and assumed GPUs interconnect usage.

| Network | Placement | Hops | Gain |
|---|---|---|---|
| FatTree | RR | 3656.54±23.57 | |
| | Greedy | 3011.46±37.52 | 21.4% |
| | ILP | 3195.22±16.80 | 14.4% |
| | ILPLoad | 2773.21±128.52 | 31.9% |
| Dragonfly | RR | 2762.66±17.22 | |
| | Greedy | 2350.61±24.85 | 17.5% |
| | ILP | 2461.85±7.21 | 12.2% |
| | ILPLoad | 2184.53±84.14 | 26.5% |
| FatTree Sparse | RR | 3903.67±31.77 | |
| | Greedy | 3027.46±37.52 | 28.9% |
| | ILP | 3224.80±14.44 | 21.1% |
| | ILPLoad | 2805.65±127.95 | 39.1% |
| Dragonfly Sparse | RR | 2762.66±17.22 | |
| | Greedy | 2350.61±24.85 | 17.5% |
| | ILP | 2460.62±13.97 | 12.3% |
| | ILPLoad | 2184.29±84.38 | 26.5% |

## A.2 EVALUATION SETUP

We run the experiments and provide measurements (including runtime in Table 1) on a machine equipped with a 128-core Intel(R) Xeon(R) Platinum 8358 CPU @ 2.60GHz CPU and 512Gb DDR3 RAM. All experiments are conducted with the Python 3 programming language. For ILP and ILP Load algorithms, the CVXPy library Diamond & Boyd (2016); Agrawal et al. (2018) was used.

Table 6: Test configuration parameters for each experiment YAML file.

| Model | R1 | R1 | R1 | 16b |
|---|---|---|---|---|
| $C_{\text{layer}}$ | 1 | 4 | 8 | 1 |
| $L$ | 58 | 58 | 58 | 27 |
| $E$ | 256 | 256 | 256 | 64 |
| $S$ | 256 | 256 | 256 | 32 |
| $C_{\text{exp}}$ | 64 | 64 | 64 | 54 |
| num_nodes_per_leaf | 4 | 4 | 4 | 1 |
| num_gpus_per_server | 4 | 4 | 4 | 1 |

## A.3 EXPERIMENTS DESCRIPTION

In this section, we list values of hyperparameters used in our experiments. There are two experiments for the DeepSeek 16b model to test the scalability of the ILPLoad approach, using real experts' activation statistics with an artificial setup of 1 GPU per server and 1 server per rack. This totally results in 64 racks and a broad network topology.

Main experiments are performed on a real-world model and network setups. For model setup, we collect loaded experts from each layer for 19529 tokens from the OASST1 dataset Köpf et al. (2023). They are extracted from 150 entries of the dataset. Then we split activations on test and train:

1. Train: 13838 activation tokens from 100 dialogs.

2. Test: 5691 activation tokens from 50 dialogs.

Then, we use the train data split only for the ILPLoad objective to obtain load-aware placements. In the evaluation phase, all algorithms are compared to each other using a test data split. Table 6 provides parameters for individual runs.

