# OpenReview forum: "Cluster Topology‑Driven Placement of Experts Reduces Network Traffic in MoE Inference"
_ICLR.cc/2026/Conference — Submitted to ICLR 2026_

### Official Review · Reviewer_rijJ · 2025-10-29

**Soundness:** 3
**Presentation:** 3
**Contribution:** 3
**Rating:** 6
**Confidence:** 5

**Summary:**

This paper addresses the important problem of efficiently deploying MoE models across distributed clusters by optimizing expert placement to minimize network communication overhead. The authors formulate the placement problem as an Integer Linear Program (ILP) that considers both network topology and expert load statistics. Experiments on DeepSeek-MoE 16B and DeepSeek-R1 671B models demonstrate improvements over baseline methods across multiple network topologies (FatTree, Dragonfly).

**Strengths:**

* The paper tackles a genuine bottleneck in deploying large-scale MoE models. With all-to-all communication consuming significant part of runtime, this is a critical issue for practitioners.

* The ILP formulation is clear and well-presented

* Testing on 4 different topologies (FatTree, Dragonfly, and their sparse variants) with visualizations provides good coverage of real-world scenarios.

* Unlike MOETuner which times out after 12 hours, the proposed ILP solves in tens of minutes, making it practically deployable.

**Weaknesses:**

* The assumption about communication could be improved. Specifically, the token replication on a same device for different experts could be reduced [1].

* One limitation is that all experiments report "network hops" as a proxy metric, but no actual wall-clock runtime measurements are provided on real clusters.

* I was wondering how sensitive is the proposed method to the profiling dataset? Does OASST1 generalize well to various domains?


[1] Luo, S., Li, P., Peng, J., Wang, H., Cheng, Y. and Chen, T., 2025. Occult: Optimizing Collaborative Communication across Experts for Accelerated Parallel MoE Training and Inference. arXiv preprint arXiv:2505.13345.

**Questions:**

See above

---

> ### Author Response · Authors · 2025-11-23
>
> Dear reviewer,
>
> Thank you for the positive evaluation of our work and relevant comments. Below, we respond to the mentioned weaknesses.
>
> > The assumption about communication could be improved. Specifically, the token replication on a same device for different experts could be reduced [1].
>
> Thank you for the suggestion! We added such a regime into our validation code. Preliminary results show that the total number of hops across all experiments decreases, but the percentage delta remains nearly the same. This suggests what co-placements can be further explored. We tried to add this constraint to our equation. For now, if we linearize the resulting equation to fit the ILP, the optimization takes more than an hour (we did not obtain the solution for this approach, even after an hour of runtime). We will explore nonlinear optimizers for co-placements settings in further work.
>
> > One limitation is that all experiments report "network hops" as a proxy metric, but no actual wall-clock runtime measurements are provided on real clusters.
>
> The number of network hops used as an objective is a relevant proxy metric for the throughput in the high-load regime, which is the primary use case for our approach.
>
> > I was wondering how sensitive is the proposed method to the profiling dataset? Does OASST1 generalize well to various domains?
>
> We perform additional experiments with datasets alpaca and vicuna. We run our approach based on statistics from the one dataset, save the obtained placement of experts, and evaluate the number of hops for the remaining two datasets based on this placement. The resulting performance is shown in Section 5.4, Table 4. This table demonstrates that the decrease in performance is 1-2%. We expect the expert placement update to be performed online if a significant shift in the experts’ statistics is observed. The potential approaches to this update include using a heuristic method to adjust the optimal solution from the previous run of ILPLoad, re-running ILPLoad from scratch on the updated statistics, and re-running ILPLoad on the updated statistics with a proper initialization (warm start approach).

---

### Official Review · Reviewer_SPx7 · 2025-10-31

**Soundness:** 2
**Presentation:** 3
**Contribution:** 2
**Rating:** 4
**Confidence:** 3

**Summary:**

This work focuses on how to efficiently deploy large Mixture-of-Expert (MoE) models for inference by jointly considering the network topology in distributed environments and the workload imbalance across experts. Two variants of integer linear programming (ILP) problems are formulated to minimize the network traffic (measured as the number of network hops) by deducing the expert placement.

**Strengths:**

1. The problem formulation is clear and the paper is generally easy to follow.
2. Unlike previous works, this paper jointly considers multiple network topologies and the workload imbalance across experts.
3. The evaluation is conducted at a large scale (671B model over 256 GPUs).

**Weaknesses:**

1. The formulation focuses on the “distance” (the length of the shortest path during network transmission). However, it is not a good proxy for network transmission cost, since different connections would have divergent bandwidths, and the network latency should also be considered (especially for small transmission messages).

2. According to Table 1, it is time-consuming for the problem solving of both ILP and ILPLoad. In practice, the request distribution may shift across time, which also affects the routing distribution. Given the long problem solving time, I’m afraid it can hardly get real-world deployment.

3. The evaluation metric is the number of network hops, rather than widely used metrics like latency, throughput, and SLO.

4. DeepSeek R1 adopts shared experts. It seems that the current work fails to take this factor into account.

**Questions:**

1. Can you provide metrics like latency, throughput, and SLO? If not, please explain why, and please clarify how the reduction (e.g., 20%) in hops can be translated to a meaningful improvement in end-to-end performance. Besides, the interconnected bandwidths should be specified.

2. How can this work be extended to dynamic routing distributions and models with shared experts?

---

> ### Author Response · Authors · 2025-11-23
>
> Dear reviewer,
>
> We thank you for the constructive feedback and the positive evaluation of the selected models for experiments and network topologies with imbalanced loading of experts. Below, we answer questions and comment on weaknesses.
>
> **W1.** Our high-level approach based on “distance” includes the mentioned granularities and effects from bandwidths and the network latency. These quantities can be easily included in our setting with proper initialization of the cost vector. However, since we do not have access to a large cluster for testing our approach, we use only vanilla initialization. In the deployment stage, our approach can include the recommended network characteristics and provide the corresponding optimal solution. So, our proxy does not limit our approach while demonstrating its end-to-end usage.
>
> **W2.** We perform additional experiments with datasets alpaca and vicuna to show that our algorithm is robust to dataset shift. We run our approach using statistics from a single dataset, save the resulting expert placements, and evaluate the number of hops for the remaining two datasets based on these placements. The resulting performance is shown in Section 5.4, Table 4. This table demonstrates that the decrease in performance is 1-2%. Based on our experiments, one could rebalance periodically in response to observed distribution shifts and, if necessary, use dynamic heuristic balancing to handle accident load spikes.
>
> **W3.** The number of network hops used as an objective is a relevant proxy metric for the throughput in the high-load regime, which is the primary use case for our approach.
>
> **W4.** The load for shared experts is proportional to the overall load and does not depend on the routers’ outputs. According to the tech report, DeepSeek uses separate GPUs for shared experts. So, we have excluded shared experts from the placement problem.
>
> **Q1.** Basically, a reduction in hops leads to locality of communication. In modern hierarchical topologies, higher-throughput links are used more often than lower-throughput links (e.g., NVLink vs network communications, leaf-switch connections vs inter-rack communications).
>
> **Q2.** Static and dynamic placements are complementary, and one can use both approaches. One can use our approach for **static** distribution and then use a **dynamic** approach to adjust expert placement to changes in the distribution of incoming traffic.
>
> We hope you find our comments sufficiently convincing and kindly ask you to increase the score for our work.

---

> > ### Comment · Reviewer_SPx7 · 2025-11-26
> >
> > > However, since we do not have access to a large cluster for testing our approach, we use only vanilla initialization.
> >
> > Did you conduct experiments with a real cluster or based on some simulators?
> >
> > > Based on our experiments, one could rebalance periodically in response to observed distribution shifts and, if necessary, use dynamic heuristic balancing to handle accident load spikes.
> >
> > The runtime of ILP and ILPLoad in Table are over 1000 seconds. I'd say this is longer than the period of load spikes.
> >
> > > The number of network hops used as an objective is a relevant proxy metric for the throughput in the high-load regime, which is the primary use case for our approach.
> >
> > Latency metrics are also essential and must be considered.
> >
> > > According to the tech report, DeepSeek uses separate GPUs for shared experts. So, we have excluded shared experts from the placement problem.
> >
> > Considering that time balance should be achieved across shared experts and the other experts, whether it affects your problem formulation and solving is unclear.

---

> > > ### Author Response · Authors · 2025-11-28
> > >
> > > > Did you conduct experiments with a real cluster or based on some simulators?
> > >
> > > Our results are obtained with a hop-based simulator. We will state this explicitly and position real-cluster or ns-3–like validation as future work.
> > >
> > > > The runtime of ILP and ILPLoad in Table are over 1000 seconds. I'd say this is longer than the period of load spikes.
> > >
> > > We agree ILP/ILPLoad runtimes are too long for short spikes. Our intent is initial and slow-timescale rebalancing (30 minutes/hours/days, to adapt to major distribution’s drifts), while for fast reactions to spikes, one can use existing rebalancing techniques, thoroughly explored in works such as Lynx [1] and Dynamic Mixture of Experts [2]
> > >
> > > > Latency metrics are also essential and must be considered.
> > >
> > > We agree that latency/SLO metrics are essential and are not measured here. Our current claim is limited to reducing traffic (hops) in the high-load regime, where throughput is the dominant bottleneck (as we need to achieve good unit economy), and we use hops as a proxy for locality. End-to-end latency/bandwidth evaluation on a more detailed (e.g., ns-3–based) simulator is ongoing work, and our ILP can directly replace hop count with latency- or bandwidth-weighted link costs.
> > >
> > > > Considering that time balance should be achieved across shared experts and the other experts, whether it affects your problem formulation and solving is unclear.
> > >
> > > As shared expert is activated every time - you just place it closest to linear layers and then solve ILP, it will be an optimal solution, as other experts in real models are activated with probability $\ll 1$
> > >
> > > [1] Gupta V. et al. Lynx: Enabling efficient moe inference through dynamic batch-aware expert selection (https://arxiv.org/abs/2411.08982)
> > >
> > > [2] Guo Y. et al. Dynamic mixture of experts: An auto-tuning approach for efficient transformer models (https://arxiv.org/abs/2405.14297)

---

### Official Review · Reviewer_r42w · 2025-11-01

**Soundness:** 3
**Presentation:** 3
**Contribution:** 3
**Rating:** 6
**Confidence:** 5

**Summary:**

This paper addresses the efficient distributed inference of large-scale Mixture-of-Experts (MoE) language models, which, despite their massive parameter counts, activate only a sparse subset of experts per token, leading to highly imbalanced expert utilization and significant network communication bottlenecks across multi-GPU clusters. The authors formalize the expert-placement problem as a topology-aware integer linear program (ILP) that minimizes the expected number of network hops required to dispatch and collect expert outputs. The model represents the cluster as an undirected weighted graph whose vertices are GPUs/servers and whose edge weights reflect inter-server distances; zero weights model intra-server NVLinks. The ILP decides, for every expert in every MoE layer, which server it should reside on, subject to per-server caps on total experts and on experts from any single layer. A load-aware extension (ILPLoad) incorporates empirical expert-activation frequencies derived from the OASST1 dataset, weighting the hop count by the probability that each expert is actually activated. Extensive experiments on two model families (DeepSeek-MoE 16 B and DeepSeek-R1 671 B) and four cluster topologies (Fat-Tree, Dragonfly, and their sparse variants) with 256 GPUs show that ILPLoad reduces average network hops by 6–30 % relative to strong baselines (round-robin and greedy placement), while remaining solvable in under 25 min for the largest configuration. The study thereby demonstrates that co-designing model mapping with network topology and workload statistics yields substantial reductions in inference-time communication overhead, improving cluster utilization and response latency for production MoE services.

**Strengths:**

1. The topic of this research is very interesting and practical: It targets the network-communication bottleneck that arises when serving Mixture-of-Experts models across a multi-GPU cluster at inference time. It proposes an expert-placement strategy that explicitly exploits the cluster’s physical topology—an issue that, while highly practical for today’s large-model deployments, has received little prior attention. Most existing studies concentrate on training-time optimizations or on balancing expert load; in contrast, this work zeroes in on reducing inference-stage network traffic, delivering clear engineering value and research novelty.
2. This paper proposes a rational approach to formalize the expert-placement task as an integer-linear-programming (ILP) problem with a clear objective and concise constraints, demonstrating strong theoretical rigor. By incorporating expert-load statistics (ILPLoad) as prior knowledge to refine the objective, they also show a deep understanding of real-world deployment scenarios.
3. Systematic experiments were conducted across multiple network topologies (Fat-Tree, Dragonfly, etc.) and two MoE models of different scales (16B and 671B), demonstrating the effectiveness of the ILP-based method in diverse scenarios. Compared with common heuristics such as Round-Robin and Greedy, ILPLoad shows significant advantages in reducing the number of network hops, achieving gains of up to 30% or more.

**Weaknesses:**

1. Scalability Concerns: While the ILP approach provides optimal solutions, the computational time (1185.9 seconds for ILP, 1397.5 seconds for ILPLoad) may not be feasible for very large-scale deployments in real-time environments. While the authors acknowledge this limitation, further investigation into methods for speeding up the optimization process or approximating solutions could be beneficial. Just a reminder: as an open-source solver, the CVXPy library is built on CBC (COIN-OR Branch-and-Cut). Its performance and efficiency can be significantly lower than other commercial solvers such as Gurobi or CPLEX.
2. Optimization Problem Complexity: While the ILP provides optimal solutions, the complexity of solving large ILP problems for extremely large models, such as Kimi K2, may pose challenges for deployment. Further discussion on how the method can be adapted or scaled for even larger models, or on how to handle more complex expert routing, would be valuable.

**Questions:**

1. Expert Load Estimation: The ILPLoad approach uses expert load statistics from the OASST1 dataset for optimizing expert placement. How robust is this approach when expert load statistics vary between different datasets or in production environments? Is there a way to dynamically update or estimate these statistics in real-time during inference?
2. Comparison to Other Methods: You mention that the ILP-based method outperforms MoETuner in terms of runtime and network traffic reduction. Can you elaborate more on the differences in approach between MoETuner and your method? Specifically, what are the key advantages of using ILP over MoETuner, aside from runtime and network traffic?

---

> ### Author Response · Authors · 2025-11-23
>
> Dear reviewer,
>
> We thank you for the positive evaluation of our work ("rational approach to formalize the expert-placement task" and "ILPLoad shows significant advantages in reducing the number of network hops, achieving gains of up to 30% or more") and answer your questions below.
>
> **W1.** We demonstrate in the presented work that even an open-source framework for solving the ILP problems of the considered dimension can solve the ILP and ILPLoad problems in a reasonable time. Since solving such a problem is done once before deployment, the reported runtime is not a bottleneck for models of the considered sizes (DeepSeek R1 has 56 MoE layers and 256 experts). The scaling of the proposed approach to even larger LLMs is the topic of future work.
>
> **W2.** First, the deployment of the resulting placement is done once for a period of time, and it is not necessary to recompute from scratch too often.
> Second, we report the runtime for the DeepSeek R1 model, which corresponds to 3801088 binary variables. At the same time, the mentioned Kimi K2 model corresponds to 5996544 binary variables. We expect that such growth still yields a feasible runtime, which can also be reduced with a warm-start approach based on the previously obtained solution or a heuristic (RR or Greedy) result. We will add such a discussion in the corresponding Section.
>
> **Q1.** We perform additional experiments with datasets alpaca and vicuna. We run our approach based on statistics from the one dataset, save the obtained placement of experts, and evaluate the number of hops for the remaining two datasets based on this placement. The resulting performance is shown in Section 5.4, Table 4. This table demonstrates that the decrease in performance is 1-2%. We expect the expert placement update to be performed online if a significant shift in the experts’ statistics is observed. The potential approaches to this update include using a heuristic method to adjust the optimal solution from the previous run of ILPLoad, re-running ILPLoad from scratch on the updated statistics, and re-running ILPLoad on the updated statistics with a proper initialization (warm start approach).
>
> **Q2.** MoETuner formulates an ILP for expert placement but considers only one or two servers and assumes a complete graph network, which does not capture real-world multi-rack topologies; moreover, extending that objective to a cluster-scale ILP becomes intractable, as we show in Table 1.
> In contrast, our formulation is explicitly topology-aware and scales to 256 GPUs and multiple racks for Fat-Tree, Dragonfly, and their sparse variants.

---

### Official Review · Reviewer_LPV4 · 2025-11-01

**Soundness:** 2
**Presentation:** 2
**Contribution:** 2
**Rating:** 2
**Confidence:** 4

**Summary:**

This paper focuses on the efficient deployment of MoE model during inference, and proposes a expert placement strategy using topology-aware modeling and ILP to minimize transmission while addressing expert load imbalance.
Experiments show the strategy achieves lower network traffic than competitors on DeepSeek models.

**Strengths:**

1. The manuscript is well-structured and easy to follow.
2. The experimental results demonstrate significant benefits.

**Weaknesses:**

1. This paper is best suited for submission to conferences focused on machine learning systems and computer architecture, as its contributions are closely related to cluster network design and optimization.
2. The target scenario and underlying motivation require further elaboration. Additionally, the analysis and evaluation do not sufficiently demonstrate the impact of key factors such as data volume, network bandwidth, and latency.
3. The paper lacks end-to-end evaluations and comprehensive comparisons with existing methods.
4. Considering only scale-out networks is insufficient to address the challenges of expert placement in practical MoE deployments, as this issue has already been thoroughly examined in existing works.

**Questions:**

Is the evaluation performed on a simulated environment or a real cluster?

---

> ### Author Response · Authors · 2025-11-23
>
> Dear reviewer,
>
> We thank you for the positive evaluation of our experimental results and presentation quality. Below, we address the questions and weaknesses.
>
> **W1.** Our study addresses the problem of deploying the Mixture-of-Experts models, which demonstrate the SOTA results of the NLP tasks. Our approach is crucial for applying MoE models to specific services and requires non-trivial ILP techniques. ICLR 2026 has a specific track devoted to Mixture-of-Experts models, with no particular limitations on the techniques used (optimization) or subdomains (cluster network design).
>
> **W2.** We propose the approach based on the explicit optimization problem that aims to minimize the traffic load from properly placing the expert layers. We demonstrate that our approach reduces traffic load, thereby minimizing redundant network communications. If the bottleneck in request processing is network throughput, then our approach can increase the number of requests processed per target period.
>
> First, we do not consider dependence on data volume, since we focus on the high-load regime of the considered cluster, i.e., we use a sufficiently large number of processed tokens to observe the gain from proper placement of experts. The low-load regime is not valuable to industry, since industrial applications generate a large number of requests per GPU.
>
> Second, network bandwidth can be incorporated into our approach via the edge weights in the cluster graph. We consider hierarchical computational clusters and assume that GPUs on the same node have an order of magnitude larger interconnection bandwidth. Since we do not have access to a sufficiently large cluster, we cannot explicitly measure and include network bandwidths in our experiments.
>
> Third, under high-load conditions, the latency for an individual request is approximately the inverse of the request-processing bandwidth. While there are inherent trade-offs between latency and throughput (e.g., via batch size adjustments), these are complementary to the problem considered in our work. Our focus is on optimizing token-volume processing, which remains independent of batch size tuning or latency/throughput trade-offs.
>
> **W3.** Our study compares the approach with heuristic competitors in Tables 2 and 3. The MoETuner competitor appears infeasible for the considered model size (see Table 1). We evaluate the considered approaches based on the introduced objective function and the runtime required to compute the target expert placement.
>
> **W4.** We do not restrict ourselves to “scale-out only” networks. Our formulation operates on a GPU-level topology graph where vertices are GPUs and edges encode both intra- and inter-server connections, with arbitrary edge weights. As described in Sec. 3.1, GPUs on the same server are connected by edges of (near) zero cost to reflect the much higher bandwidth of NVLink/NVSwitch compared to inter-server links.
>
> This choice is not a limitation of the method, but a parameterization. By changing these intra-server edge weights, our ILP can model any scale-up topology (rings, NVSwitch crossbars, etc.) in combination with the scale-out fabric.
>
> We agree that expert placement and MoE serving have attracted increasing attention. However, to the best of our knowledge, prior work falls into two different categories:
> 1. Topology-agnostic or small-scale placements, such as MoETuner, formulate an ILP for expert placement; however, they consider only one or two servers and assume a complete graph network, which does not capture realistic multi-rack topologies.
> 2. Dynamic/on-the-fly rebalancing and routing like Lynx, expert sharding, and dynamic gating focus on runtime expert replication, batch-aware routing, and gating changes to adapt to online load imbalance. These methods typically assume a homogeneous or oversimplified view of the network fabric and optimize steady-state load rather than minimizing communication under a concrete cluster topology.
>
> Our contribution complements dynamic approaches, as we present a static, offline (or recalculated in the background) method that guarantees the optimality of the expert placement under the target cluster network. Dynamic optimization can still be used between static placements to adjust placement for the accident load spike.

---

> > ### Comment · Reviewer_LPV4 · 2025-11-27
> >
> > You frequently emphasize that your methods are adaptable to various scenarios, yet your paper lacks experimental evidence to substantiate these claims. From a systems research standpoint, your experiments appear insufficient to validate your problem modeling and solution approaches. Despite not having access to a large cluster, you can still perform experiments using well-recognized simulators. Additionally, it is crucial to specify the software/hardware platform and configurations used in your experiments within the paper.
> >
> > Your focus is on inference scenarios rather than training, where the load of MoE can be affected by numerous practical factors, such as memory-bound decoding, varying sequence lengths of requests, and inference system designs (with or without PD and AE disaggregation). This is an issue that cannot be dismissed as "not valuable to industry." You need to thoroughly verify your approach, delineate the optimization space, and address any negative impacts that may occur when operating outside of this space.

---

> > > ### Author Response · Authors · 2025-11-28
> > >
> > > We agree that, from a systems perspective, the scope and assumptions must be clear.
> > > Our work targets static, topology-aware MoE expert placement in the high-load, network-bound inference regime; it is explicitly orthogonal to the design of the inference stack (PD/AE disaggregation, batching policy, or kernel-level optimizations). These factors only change (i) link costs and (ii) the expert-activation statistics, both of which are parameters in our ILP; once profiled, the same formulation applies.
> > >
> > > We agree on the steps needed to proceed with hardware simulations. Specifically, we are now working to extend our ns-3 simulation with parallelism (the single-threaded version is slow). Currently, we are exploring complementary parallelism techniques for linear layers in the simulator. Nevertheless, hop measurement is a good proxy for locality, isolating other factors.

---

### Meta-Review · Area_Chair_RmpS · 2026-01-13

**Summary:**

**Summary**

This paper addresses the important problem of efficiently deploying MoE models across distributed clusters by optimizing expert placement to minimize network communication overhead. The authors formulate the placement problem as an Integer Linear Program (ILP) that considers both network topology and expert load statistics. Experiments on DeepSeek-MoE 16B and DeepSeek-R1 671B models demonstrate improvements over baseline methods across multiple network topologies (FatTree, Dragonfly).


**Strengths**
- **Expert-Placement Strategy**: The paper proposes a novel expert-placement strategy that leverages the physical topology of multi-GPU clusters to address network-communication bottlenecks during inference for Mixture-of-Experts models, focusing on reducing network traffic rather than training-time optimizations.

- **Integer-Linear-Programming (ILP) Formulation**: It formalizes the expert-placement task as an ILP problem, incorporating expert-load statistics to enhance the objective function, showcasing theoretical rigor and a practical understanding of deployment scenarios.

- **Extensive Evaluation and Performance Gains**: The manuscript presents systematic experiments across various network topologies and large-scale models, demonstrating that the ILP-based method significantly reduces network hops and outperforms common heuristics, achieving up to 30% improvements in efficiency.

**Concerns**
- **Target Scenario and Motivation**: The paper requires further elaboration on the target scenario and underlying motivation for the research, as it currently lacks clarity.

- **Evaluation and Comparisons**: There is a lack of end-to-end evaluations and comprehensive comparisons with existing methods, which diminishes the demonstration of the impact of key factors like data volume, network bandwidth, and latency.

- **Scalability and Optimization Concerns**: The ILP approach, while optimal, has significant computational time that may not be feasible for large-scale deployments. Further investigation into speeding up the optimization process or approximating solutions is necessary.

- **Evaluation Metrics and Assumptions**: The paper relies on "network hops" as an evaluation metric instead of more relevant metrics like latency and throughput, and assumptions about communication and expert routing could be improved for better accuracy and applicability.

**Decision**
My recommendation is to enhance the paper by incorporating additional experimental data and metrics, as suggested by the reviewers. Although your focus on adaptability is commendable, the absence of experimental evidence weakens your claims. The limited validation of your problem modeling and solution approaches raises concerns from a systems research standpoint. Even without access to a large cluster, it is crucial to use established simulators to effectively demonstrate your methods.

**Reviewer Concerns:**

The author's feedback addressed most concerns, but the validation of the approach remains a weak point. The reviewer recommended incorporating large-scale simulations and additional metrics to strengthen this aspect.

**Reviewer Scores:**

As it is mentioned above, the reviewer would not changed their score.

---

### Decision · Program_Chairs · 2026-01-26

Reject